# Role of Fork-Head Box Genes in Breast Cancer: From Drug Resistance to Therapeutic Targets

**DOI:** 10.3390/biomedicines11082159

**Published:** 2023-08-01

**Authors:** Ali Hazazi, Samia S. Alkhalil, Ahad Amer Alsaiari, Amal F. Gharib, Hayaa M. Alhuthali, Shanika Rana, Abdulaziz A. Aloliqi, Alaa Abdulaziz Eisa, Mohammad Raghibul Hasan, Kapil Dev

**Affiliations:** 1Department of Biotechnology, Jamia Millia Islamia, New Delhi 110025, India; sadaf924@gmail.com; 2Department of Pathology and Laboratory Medicine, Security Forces Hospital Program, Riyadh 11481, Saudi Arabia; ahazazi@sfh.med.sa; 3Department of Clinical Laboratory Sciences, College of Applied Medical Sciences, Shaqra University, Alquwayiyah 11961, Saudi Arabia; salkhalil@su.edu.sa; 4Department of Clinical Laboratory Sciences, College of Applied Medical Sciences, Taif University, Taif 21944, Saudi Arabia; ahadamer@tu.edu.sa (A.A.A.); amgharib@tu.edu.sa (A.F.G.); hmhuthali@tu.edu.sa (H.M.A.); 5School of Biosciences, Apeejay Stya University, Gurugram 122003, India; ranashanika29@gmail.com; 6Department of Medical Biotechnology, College of Applied Medical Sciences, Qassim University, Buraydah 52571, Saudi Arabia; aaalieky@qu.edu.sa; 7Department of Medical Laboratories Technology, College of Applied Medical Sciences, Taibah University, Medina 30002, Saudi Arabia; aeisa@taibahu.edu.sa; 8Department of Clinical Laboratory Sciences, College of Applied Medical Sciences, Shaqra University, Riyadh 15342, Saudi Arabia

**Keywords:** forkhead genes, forkhead proteins, *FOXA*, *FOXM1*, *FOXC*, *FOXO*, *FOXP*, miRNAs, drug resistance, stem cells, breast cancer

## Abstract

Breast cancer has been acknowledged as one of the most notorious cancers, responsible for millions of deaths around the globe. Understanding the various factors, genetic mutations, comprehensive pathways, etc., that are involved in the development of breast cancer and how these affect the development of the disease is very important for improving and revitalizing the treatment of this global health issue. The forkhead-box gene family, comprising 19 subfamilies, is known to have a significant impact on the growth and progression of this cancer. The article looks into the various forkhead genes and how they play a role in different types of cancer. It also covers their impact on cancer drug resistance, interaction with microRNAs, explores their potential as targets for drug therapies, and their association with stem cells.

## 1. Introduction

Globally, breast cancer has been known to be the second biggest cause of fatalities in women, and it is also the most often diagnosed malignancy. Although technology and research, in the past few decades, have enabled better and more effective treatment, coupled with early diagnosis, a copious number of patients have been known to be resistant to current treatment approaches [1]. The disease’s molecular uniqueness, as defined by unique gene expression profiles and the overexpression of driver oncogenic proteins, has long posed a tremendous challenge in clinical therapy [2]. There are many leading causes and factors related to tumor proliferation in the breast tissue, and forkhead family proteins produced by forkhead family genes have been garnering worldwide attention for their dominant role.

Forkhead family proteins are convoluted in cell proliferation, differentiation, the regulation of cell cycle and lifespan, embryogenesis, and differentiation via coupling with the genes that govern transcription and DNA repair [3]. This family acts as an integral regulator in physiological development and cellular homeostasis and has been associated with many congenital disorders. The binding domain in a forkhead-box gene is a continuous and preserved winged helix, and the forkhead family includes many transcription factors that are extremely specific to various cell and tissue types [4]. Even though all subgroups of the forkhead family share the DBD structure known as FHD (Forkhead-DNA-binding domain), they constitute varied activation and repression zones [2,5]. FOX protein binds to different DNA sequences by recognizing specialized patterns in the DNA, which allows the unwinding of chromatin and expedites the dynamism of other regulators. The major functional domains and motifs in human FOX proteins include Zinc finger, Leucine zipper, and conserved region 1. The size of FOX-DBD is nearly 100 amino acids in length, having a highly conserved N-terminal portion, while the c-termini residue lacks conservation to much extent [2]. 

Based on protein sequence homology, there are approximately 50 genes that encode forkhead proteins in humans, and they have been divided into 19 subclasses since the discovery of the first FOX gene (FOXA to FOXS) [2,6]. Despite the fact that they conserve motif sequence homology, they have been shown to differ in a variety of functions, i.e., cell cycle progression, differentiation, apoptosis, and metabolism [2]. Under varied oncogenic circumstances, FOX transcription factors play a variety of opposing roles with unusually high biological specificity. Several members of the family have been shown to be either tumor-suppressive or -promotive, depending on their mode of interaction with different transcriptional pathways at different stages [2,7]. In general, FOX proteins affect angiogenesis, DNA damage repair, metabolism, cell destiny, angiogenesis, cell proliferation and differentiation, metabolism, and senescence [8].

The onset, invasion, development, and medication resistance of cancer are all correlated with the dysregulation of FOX proteins. They can also control other cancer-related pathways that help cells survive in difficult circumstances. For instance, the FOXO subfamily activates antioxidant enzymes in response to cellular stress to shield the cell from oxidative damage [9]. Among the 50 FOX-encoding genes categorized into 19 subfamilies, FOXM1, FOXA, FOXO, FOXP, and FOXC have garnered the most attention, especially from oncologists [10]. 

Breast cancers are often divided into ER-negative and ER-positive subtypes [11]. The diagnosis of patients with ER-positive breast tumors is typically better, in part because of how well they react to hormonal treatment. Contemporary gene expression profiling studies have enabled the classification of breast tumors into five intrinsic subgroups with varying prognostic significance [12]. These subtypes comprise luminal types [A and B], ER-positive/HER2-negative, basal type, and normal-like breast cancers. Luminal type [A and B] breast cancer express estrogen, with luminal type A have increased estrogen levels, and they have a better prognosis [13].

With extremely complex and extensive networks, the *FOX* family participates in the development, upkeep, advancement, and metastasis of cancer at many regulatory levels (Table 1). Major elements of the characteristics that distinguish cancer are also connected to FOX proteins. This review article describes the different *FOX* genes and their roles in primarily breast cancer proliferation and progression, different transcriptional factors pertaining to the genes, applications in microRNAs, and drug resistance. 

## 2. Role of Various Forkhead Boxes in Tumor Proliferation and Progression

### 2.1. Forkhead Box A in Cancer

Pioneer factor forkhead box genes *FOXA1/A2/A3* are momentous for the development of endoderm and organs originating from the endoderm. As pioneer factors, they facilitate chromatin access for other transcription factors so that they can carry out their tissue-specific tasks [14,15]. Based on their numerous actions, particularly the apropos triggering of cellular invasion in nearby and outlying tissues and metastasis, DNA damage and mutation, and persistent cell cycle signaling, *FOXA1* and *FOXA2* do play significant functions in carcinogenesis. By being additionally linked to several malignancies, *FOXA1* and *FOXA2* exhibit tumor-type-specific characteristics that depend on specific transcriptome connections [16]. Lung, prostate, and esophageal cancers are all highly linked with the presence of *FOXA1* upregulation [17]. In bladder cancer, the diminished expression of the *FOXA* gene has been associated with muscle-invasive tumor progression, which then advances to the metastatic phase [18].

The two main cancer malignancies that are affected by hormones are breast and prostate cancer. FOXA demonstrates a crucial role in the synchronization of estrogen-receptor and androgen-receptor functions, so it is not as startling to discover the presence of forkhead-box A gene expression in ER-positive breast cancer and AR-positive prostate tumors. *FOXA1* is expressed in a small fraction of ER-negative breast tumors that express AR [19]. In breast cancer cell lines, the *FOXA1* gene is fundamentally manifested when cells express ER. Furthermore, cDNA microarray cluster result analysis associates *FOXA1* gene transcripts with other genes, such as ER, GATA-3, and X-box binding protein [XBP-1], and this is a characteristic feature of the luminal subtype A of breast cancer [20,21]. 

The ER-estrogen-FOXA1-GATA-3 axis has a determining function in the beginning and/or progression of luminal-type tumors, and this has recently been proposed by the attestation of *FOXA1* as a pioneer factor, essential for the expression of the bulk of estrogen-inducible genes, like a breast tumor (Figure 1) [22,23]. As luminal type A breast cancer tumors require estrogen for survival and growth, such patients respond better to estrogen therapy, whereas patients with luminal type B breast cancer tumors have a meager prognosis. Epidermal growth factor receptor (EGFR) and human epidermal growth factor receptor (HER2) are two examples of growth factor receptors that can mediate redundant survival and proliferation pathways in luminal type B malignancies [24]. 

Thus, *FOXA1* has been commendatory with a luminal subtype and favorable prognosis. As *FOXA* is a pioneer factor, it becomes easy for estrogen to target the regulatory regions in target genes in healthy breast tissue [25]. Additionally, there appears to be a positive regulatory loop between the *FOXA1* and the target genes, and changes in this pathway may result in the cells undergoing a malignant transformation. Prior research demonstrates the significance of *FOXA1* expression for a favorable prognosis in steroid-dependent carcinomas, like breast and prostate cancers [26]. In basal-like (or triple-negative) breast cancer cell lines, the in vitro suppression of *FOXA2* gene expression decreases proliferation and mammosphere development. *FOXA2* is connected to more aggressive tumor behavior because it plays a part in the mechanisms of proliferation [27]. In light of these encouraging findings, *FOXA2* is supported as a peculiar remedial target for the nursing of a particular category of people with triple-negative/basal-like breast cancer who are at an increased danger of recurrent cancer [28].

### 2.2. Forkhead Box M1 in Cancer

*FOXM1* is a predominant modulator of cancer growth and metastasis that is selectively expressed in dividing cells [29]. The higher expression of *FOXM1* is linked to lower patient survival, while the overproduction of the *FOXM1* gene transcript is prevalent in breast cancer [30]. *FOXM1* helps cancer cells evade growth suppressors by turning on cell-cycle regulators and anti-oxidant genes and the advancement along the EMT composition, annexation, and the evolution of pre-metastatic niches [30,31,32]. Fascinatingly, inhibiting *FOXM1* alone is thought to be sufficient for addressing multiple cancer processes [33]. 

*FOXM1* is so ubiquitous in breast cancer that all subtypes produce the *FOXM1* gene transcript that also happens to be indispensable for treatment resistance, the transition of epithelial cells to mesenchymal cells (EMT), annexation, and metastasis [34]. *FOXM1* pronouncement fluctuates with the cell cycle phase under normal physiological circumstances, rising during the S phase, and is the most pronounced in the G2-M phase. In pre-symptomatic types of breast cancer, *FOXM1* is necessary for augmentation and mitosis and modulates the genes transcription factors, such as p27kip1, cyclin D1, and cdc25, that are affiliated with cell division modulation at the G1-S and G2-M transition points. As an autonomous feature of estrogen concentrations, the reduced levels of *FOXM1* abate the propagation of mammary tumor cells without alleviating cell death [35]. *FOXM1* controls breast cancer mitosis and EMT, coupled with proliferation. *FOXM1* depletion encourages polyploidy and chromosomal instability. Because of defective cytokinesis that results in centrosomal expansion and the development of multipolar spindles, stable FOXM1 knockdown may cause mitotic catastrophe.

When analyzed against customary breast tissue, the FOXM1 transcript is more abundant in breast cancers [36]. When compared to normal breast tissues (*n* = 14), fibrocystic breast tissues (*n* = 17), or fibroadenomas (*n* = 7), penetrating ductal breast carcinomas (*n* = 194) show upregulated *FOXM1* transcription [37]. Real-time PCR analysis and immunohistochemical staining both demonstrate the higher transcription levels of the *FOXM1* gene in mammary tumor tissues, as opposed to normal breast tissues [36,37].

*FOXM1* controls ER-alpha production, transitions, and interactions, as well as the transcriptional activity in luminal subtypes; forkhead binding domains have been discovered in the ESR1 (ER alpha) promoter zone [38,39]. It has been substantiated that an inverse relation exists between the *FOXM1* transcript and beta1 production magnitude, prompting that ER-beta1 appears to suppress the transcription of the *FOXM1* gene by expelling ER-alpha molecules from the *FOXM1* gene promoter site in ER-positive breast tumors [40]. In patients with luminal subtypes receiving supportive chemotherapy alone or receiving the drug tamoxifen alone, noticeably increased *FOXM1* transcript levels were observed, corresponding with a diminished outlying metastasis-free situation, without relapse, or comprehensive survival rates, suggesting a potential link to noncompliance in terms of chemotherapy or endocrine therapy [41,42] (Figure 2).

By enhancing YAP1’s transcriptional activity, FOXM1 increased the capacity of triple-negative breast cancer cells to proliferate, form clones, and migrate. Additionally, *FOXM1* preserved cell stemness by using the Hippo pathway. YAP1-TEAD binding inhibitor Verteporfin decreased the levels of OCT4 and NANOG molecules, but they were increased by XMU-MP-1. In conclusion, *FOXM1* enhanced the advancement of mammary tumors through the Hippo pathway and can be utilized as a novel approach to treating breast cancer [43].

### 2.3. Forkhead Box O in Cancer

Among the FOX proteins, the FOXO subclass is undeniably the one that attracts the most attention from researchers. Different biological processes are carried out by each FOXO protein. For instance, FOXO1 is crucial for angiogenesis, whereas FOXO3A is crucial for the development of ovarian follicles [44]. As opposed to FOXM and FOXC subclasses, which are true oncogenes, FOXO proteins have a variety of catabolic and anabolic processes at several levels. The imbalanced modulation of FOXOs may result in tumor proliferation [45]. Cellular energetics, evading growth suppressors, replicative immortality, initiating angiogenesis, genomic unpredictability and mutation, and assisting proliferative signaling are just a few of the hallmarks of cancer that FOXOs play a part in [46,47,48,49]. 

The carcinogenic functions of FOXOs in breast cancer are demonstrated by their control of numerous procedures necessary for carcinogenesis. Reassessing the functions of FOXO proteins in tumor proliferation is crucial for this reason. The majority of the post-translational modifications control FOXOs. For instance, the PI3K/Akt/Insulin (phosphorylated by phosphatidylinositol 3-kinase/RAC-serine/threonine-protein kinase) signaling pathway typically modifies FOXO members [47]. It has been demonstrated that by inhibiting Akt, ERK, and IKK via the current treatment methods, the indirect overexpression of FOXOs is enabled [48]. 

Enough research has led us to conclude that SIRT and *FOXO* genes have a predominant part in the evolution of tumors, but their functions in metastasis are still ambiguous. *SIRT1*, *p21*, *p53*, *E2F1*, and *FOXO* protein expression levels were higher in 67NR groups in primary tumors. SIRT1, E2F1, and FOXO protein expression concentrations were discovered to be elevated in metastatic tissues, but p53 and p21 expression levels were found to be demoted. The molecular role of the SIRT and FOXO proteins in the development and spread of tumors was also empirically supported by IPA analysis. Thus, FOXO proteins and SIRT1 work together to promote metastasis [50].

FOXO3A dephosphorylation, nuclear and cellular translocation, and the disruption of its interaction with SirT6 are brought on by the protracted therapy of luminal breast cancer cells with AKT inhibitors, which also causes FOXO3A acetylation coupled with BRD4 recognition. When BRD4’s BD2 domain is recognized by acetylated FOXO3A, the BRD4/RNAPII complex is attracted to the CDK6 gene promoter and causes the production of transcription factors. The in vitro and in vivo disinclination of luminal breast cancer cells to AKT inhibitors is considerably mitigated by the pharmacological suppression of either BRD4/FOXO3A association or CDK6 (Figure 3) [51]. Glucose, amino acids, and lipid metabolism are just a few of the metabolic processes that FOXOs play a role in. Consequently, FOXOs could open a wide therapeutic window for the application of metabolic disruptors [52].

### 2.4. Forkhead Box P in Cancer

A functionally varied subfamily of proteins known as the FOXP proteins is notable for its coactive participation in development during the embryonic stage, including brain development [53]. The FOXP subfamily has also been shown to have an integral role in language and speech development centers in the brain, autoimmune diseases, and cancer initiation and progression [54,55,56]. Angiogenesis, cell death resistance, maintaining proliferative signals, tumor-promoting inflammation, evading growth suppressors, and genomic instability and mutation are all factors that contribute to FOXP-dependent cancer proliferation and development (Figure 4). The ability of the FOXP subfamily to form similar and distinctive dimers with paralogs, known as FOXP1/2/4 interactions, is one of its distinctive characteristics [53]. 

FOXP2 plays a dual function in the development of cancer and oncogenesis, primarily acting as a repressor. For instance, FOXP2 can interact with CTBP1, which is a transcriptional corepressor known for controlling and targeting the production and regulation of various tumor inhibitors, such as BAX, PTEN, and p16 [57]. In epithelial cancerous cell lines, including breast cancer and lung cancer, FOXP1 functions as a tumor suppressor and is typically thought of as a transcriptional repressor. However, B-cell lymphomas overexpress FOXP1, and individuals who have increased FOXP1 expression typically have a substandard prognosis. In particular, FOXP1 is related to the proliferation of B cells throughout lymphocyte development [58]. Breast cancer, prostate cancer, and gastric cancer had better prognoses than NSCLC, colorectal cancer, and cervical cancer when FOXP3 is overexpressed [59]. 

When compared to wild-type cells, FOXP3-overexpressing MDA-MB-231 cells exhibited an enrichment of a gene signature linked to apoptosis, according to gene set enrichment analysis. Further investigation revealed that FOXP3-MDA-MB-231 cells have elevated levels of programmed cell death 4 (PDCD4), a crucial protein implicated in cell death. Both Western blotting and reverse-transcription-quantitative PCR demonstrated that FOXP3 increased the production of PDCD4 in mammary tumor cells. A public database examination of clinical samples revealed an association between PDCD4 expression levels and the symptomatic stages of breast cancer [60].

### 2.5. Forkhead Box C in Cancer

The FOXC subfamily has been shown to play a vital part in cardiovascular development [61]. In fact, it has been proven that embryos without FOXC1/FOXC2 expression manifest several abnormal cardiovascular phenotypes and even die within a few days of delivery [62]. Angiogenesis, invasion, metastasis, invasion of growth eliminators, genomic instability and mutation, and the maintenance of proliferative signals are the primary functions of FOXC1 and FOXC2 in cancer (Figure 5). Breast cancer, liver cancer, Hodgkin’s and non-lymphoma, Hodgkin’s pancreatic cancer, and endometrial cancer are only a few of the cancers that FOXC1 is linked to [63,64]. 

In a breast cancer model, a reduction in FOXC1 expression retards the development of cancer cells and converts fibroblast-like cells to epithelial cells. Furthermore, patients with basal-like breast cancer have a worse forecast, and FOXC1 is positively linked to cancer spread [65]. When FOXC1 activates the EMT process in hepatocellular carcinoma, the cancer cells are better able to migrate and invade. Patients with increased levels of FOXC1 expression typically have worse prognoses [66]. 

Like FOXC1, FOXC2 plays a significant part in the evolution of different cancer types. Breast, stomach, lung, cervical, prostate, and ovarian cancers all exhibit the overexpression of FOXC2. Notably, FOXC2 has the capacity to modulate the general upkeep of tumor cells and is involved in lipid alteration via kinases in tumor cells, whereas the role of FOXC1 in cancer metabolism is still a topic of research [67].

Thus, it is evident that the FOXC subfamily has a fundamental function in cancer development and progression, which makes it noteworthy. Further research on related cellular pathways could assist with the production of more efficient therapeutic strategies and better outcomes.

### 2.6. FOX Proteins in Cancer Drug Resistance

On a clinical level, a significant hindrance to current cancer therapies is the emergence of opposition to both established and recently developed molecular-targeted medicines [68]. It is interesting to note that *FOX* gene proteins are also related to several resistance pathways and the operations of both traditional cytotoxic chemotherapy and molecular-targeted medicines. Changes in drug targets, drug metabolism, cancerous stem cell populations, death signals, cell persistence, and drug targets all have functions in the relationships between forkhead-box proteins and the emergence of drug resistance [69]. For instance, resistance to chemotherapy and a worse mortality rate in cancer patients is strongly related to diverse FOXM1 or FOXOs protein production levels. 

In breast cancer patients, Nijmegen breakage syndrome gene-targeting by *FOXM1* can regulate senescence induced by DNA damage and epirubicin resistance [70]. *FOXM1* targets the X-lined prohibitor of the apoptosis gene (XIAP), and survivin can be a potential component for resistance in breast cancer patients [60]. Deregulating FOXM1 transcript production to control kinesin family member 20A in mitotic catastrophe can regulate paclitaxel resistance [71]. In recurrent cancers, FOXC1 production is correlated with reduced or extremely traced estrogen receptor (ER) expression, and through inhibiting GATA binding protein 3 binding, FOXC1 participates in ER silencing and has been linked to endocrine resistance [62]. It has been proposed that platelet-derived growth factor receptors are key players in the chemoresistance of breast cancer caused by FOXQ1, which could have consequences for the evolution and maturation of novel therapeutic regimens for the disease [72]. By regulating p27, FOXD1 can assist the development and chemoresistance of breast cancer [73]. By activating interleukin-6, FOXA1 is downregulated, which results in breast tumor cells that are opposed to tamoxifen, having the characteristics of cancer stem cells [74]. By prohibiting FOXC2-mediated EMT, FOXF2 may further aid the multidrug opposition of basal-like breast cancer [75].

Alternatively, the abnormal stimulation of the repair of DNA damage may be related to genotoxic treatment resistance as well as cancer progression and cancer initiation. Aiming for FOXM1 and FOXOs has the potential to treat genotoxic drug resistance because convincing data demonstrates that the FOXOs-FOXM1 forkhead transcription factor axis alters the response to DNA mutation [76,77]. A significant obstacle to contemporary cancer therapy in the clinic is the emergence of an aversion to both established prosaic medicines and recently developed molecularly targeted therapies. Intriguingly, unregulated signaling through the transcription factors FOXO3 and FOXM1 is consistently connected to the resistance procedures to “traditional” cytotoxic chemotherapies and to molecularly targeted treatment remedies. This is because FOXM1 and FOXO3 are involved in controlling the genes correlated with vital drug-action-related biological mechanisms like drug efflux, stem cell renewal, cell survival, DNA repair, and dysregulated mitosis [78]. Changes in drug metabolism, drug targets, the amount of cancer stem cells, the pace at which DNA damage is repaired, and cell survival and death signals can all have a function in the pathways for the evolution of drug resistance. Despite being distinct and varied, these procedures, which lead to the progression of drug resistance, always co-ordinate their actions and functions via the FOXO3-FOXM1 axis (Figure 6 and Figure 7).

FOXM1 has been shown to be vital in mediating resistance to genotoxic substances, such as radiation and epirubicin, through the modulation of genes associated with DNA damage repair, such as BRIP1 and NBS1. Similar to this, it has been shown that FOXM1 overexpression confers the development of cisplatin resistance in breast tumor cells. In contribution, the overproduction of FOXM1 transcript increases docetaxel’s chemoresistance in gastric cancer and may serve as a prognostic predictor [80,81,82]. In order to promote the restoration of or resistance to the DNA damage caused by genotoxic chemicals, FOXM1 overexpression or FOXO3 suppression can stimulate the functions of genes that respond to DNA damage. This helps malignant cells endure the genotoxic effects of anticancer drugs. In order to liaise mitotic cataclysm and senescence in breast cancer cells, paclitaxel can also downregulate FOXM1. The diminished expression of FOXM1 transcript is at least partially responsible for this cytotoxic function. In agreement with this, the overexpression of FOXM1 forkhead box is associated with a reduced mortality rate for breast cancer patients and can augment the tolerance of breast cancer cells to the medicine paclitaxel [83]. 

As an alternative, using the right drug cocktails to cure cancer and control drug tolerance is a well-established basis in cancer treatment remedies. Recent research has demonstrated that PI3K-Akt pathway inhibitors, like OSU-03012, have been demonstrated to improve FOXO3A dephosphorylation and nuclear translocation in breast tumor cells and can activate FOXO3A [84]. Furthermore, an additional Akt inhibitor, MK-2206, can activate and dephosphorylate FOXO3A and may work in conjunction with doxorubicin and other standard genotoxic medications to treat liver cancer [85]. Understanding the upriver modulators, downriver targets, and cell-level roles of FOX transcripts, as well as the transcriptional programs they orchestrate (in particular, cancer disease states), is crucial for determining the best possible targets for modifying FOX proteins [86].

### 2.7. Negative Regulation of FOX Proteins by miRNAs

A novel bracket of short RNAs that do not code for proteins called miRNAs is 18–25 nucleotides in length. MiRNAs are deeply rooted in several biological processes, including cell differentiation, stress resistance, and cancer, according to studies from the last 10 years. Numerous studies have shown that miRNAs in cancer patients under diverse clinical circumstances regulate FOX proteins. Recent research suggests that several malignancies, including esophageal cancer, colorectal cancer, hepatocellular carcinoma, and triple-negative breast cancer, specifically target the *FOX* genes [87,88,89,90]. For instance, miR-342 has been shown to reduce the levels of *FOXM1* and *FOXQ1* production by intuitively collaborating with the presumptive 3’-UTR binding sites of these genes, preventing colorectal cancer cells from proliferating, migrating, and invading in a xenograft animal model [87]. One of the direct targets of miR-204 is *FOXM1*, and miR-204’s functional impact on esophageal cancer cell lines depends on *FOXM1* [88]. By altering its immediate objectives, such as *FOXQ1*, *FOXG1*, and *FOXE1*, miR-422a expression was restored, which greatly reduced tumor development and liver metastasis in xenograft tumor models [91].

In HER2 (or erb2)-positive breast cancer, a humanized model of anti-HER2 monoclonal antibody (Ab) called Herceptin has proved to be a successful HER2-targeted remedy for preliminary and metastatic HER2-positive breast cancer [92]. In maintaining the fundamental IGF2/IGF-1R/IRS1 signaling in cells sensitive to Herceptin, *FOXO3A* modulates certain miRNAs to supervise IGF2 and IRS1 production [93]. The fundamental function keeps PPP3CB, a sub-unit of serine/threonine protein phosphatase 2B, expressed to prevent the phosphorylation of *FOXO3A* (p-FOXO3A), resulting in the production of miRNAs that target IGF2 and IRS1. Nevertheless, Herceptin-resistant cells have elevated p-*FOXO3A* concentrations as a result of PPP3CB transcriptional suppression, which breaks the negative feedback inhibitory arc created by *FOXO3A* and the miRNAs. This causes IGF2 and IRS1 to be upregulated [94]. The importance of a negative feedback inhibitory arc for IGF-induced signaling in cellular functions and maintenance has been demonstrated [95]. Additionally, a considerably higher level of IGF2 in the blood and IRS1 was found in the tumors of patients with breast cancer who did not respond well to Herceptin-containing regimens. 

The production of FOXM1 is upraised in breast cancer cells, and patients with breast cancer have a substandard prognosis and transient overall endurance as a result. In response to FOXM1 knockdown, Hamurcu Z, et al. used microarray technology to analyze the production outlines of 752 miRNAs in extremely militant and advanced triple-negative breast cancer (TNBC) cells. They found 13 miRNAs that showed differential expression, with 3 miRNAs stimulation and 10 miRNAs downregulated [96]. Increased FOXM1 expression is linked to diminished chances of patient mortality and induces the production of the tumor inhibitor miR200b-5p and oncomiR miR-186-5p, showing that the FOXM1/miRNA signaling pathway may be associated in the improbable prognosis of cancer patients and symbolizes a prospective medicinal target in triple-negative breast cancer [97]. 

It has been documented that miRNA-937 (miR-937) has aberrant expression in stomach and lung malignancies, where it may play tumor-inhibitor or carcinogenic functions in tumor proliferation, inclusive of the development of cancer [98,99]. MiR-937 expression levels were noticeably lower in breast cancer, and the TNM stage and lymph node metastases were substantially linked with this underexpression. The upregulation of miR-937 was suppressive of breast cancer cells’ capacity to proliferate, develop, migrate, and invade. Additionally, ectopic miR-937 expression slowed the expansion of breast cancer tumors in patients. In breast cancer, it was discovered that miR-937 directly targets the forkhead box Q1 (FOXQ1) mRNA, and FOXQ1 was shown to be overexpressed, and this overexpression exhibited a negative correlation with miR-937 expression. Thus, MiR-937 directly targets FOXQ1 mRNA to operate as a cancer inhibitor in breast tumors and slow down the growth of the disease [100] (Table 2).

When taken together, these investigations show that the modulation of the FOX protein pathways involves an additional layer of intricacy. Further study into the intricate meshwork of miRNAs and FOX transcripts will yield improved methods for enhancing tumor therapies.

### 2.8. FOX Family and Stem Cells

The consumption of embryonic stem (ES) cells in fundamental science, regenerative medicine, and drug development is a possibility. Recent research revealed a link between the FOX family and stem cells. In luminal breast cancer, there is a link between the number of carcinogenic stem cells (CSCs) and *FOXA1* expression (BC). The production of *FOXA1* transcript and the genes associated with stemness have been discovered to be more elevated in mammosphere-forming cells than in adherent culture cells [101]. A study found that when embryonic stem cells differentiate, *FOXA2* modulates nucleosome depletion and gene activation. Additionally, it was demonstrated that *FOXA3* contributed to the demarcation of ESCs into endoderm [102]. 

Furthermore, *FOXC1* is found to be highly expressed in basal-like breast cancer (BLBC). It promotes the migration and invasion of cancerous cells by activating the NF-KB signaling pathway. Hedgehog signaling has also been found to be activated in cancerous cells that have an increased production of *FOXC1* transcript. The *FOXC1* gene has also been demonstrated to be in negative correlation with the expression of estrogen receptors in cancerous cell membranes [89]. 

Maintaining blood-producing stem and progenitor cells in the myeloid tissue necessitates the maintenance of niches. By increasing CXCL12 and stem cell factor production and promoting the generation of CXC chemokine ligand (CXCL)12-abundant reticular (CAR) cells, *FOXC1* is a crucial governor of this niche establishment [103]. Additionally, it was discovered that *FOXC1* controls stem cell quiescence and conserves the chronic tissue-regenerating potential of stem cells by sustaining the hair follicle stem cell niche [104]. *FOXC1* controls early cardiomyogenesis, as well as the utilitarian characteristics of cardiomyocytes derived from ESCs [105]. *FOXC2* overexpression was ample in bringing about CSC characteristics and unprompted metastasis in mutated human breast epithelial cells, similar to *FOXC1* expression [106]. Likewise, a gene induced by forkhead-box C2 has a production trademark that was enhanced in the EMT and CSC-containing claudin-low/basal B breast cancer subtype [107].

*FOXD3* is crucial in the control of EpiSCs. Downriver of the Wnt/-catenin signaling suppressor, self-renewal is mediated by overexpression of FOXD3 [108]. As a reprogramming mediator, *FOXF1* aids in the reprogramming toward stemness, whereas *FOXF2* decreases the frequency of Lgr5+ stem cells by blocking Wnt signaling in intestinal fibroblasts [109,110]. Inhibition of Wnt/-catenin signaling promotes the growth of Epiblast stem cells. Blocking of all Wnt proteins can lead to excessive catenin retention in the cytoplasm. *FOXD3* has been shown to be a key inhibitor of IWR-1 (a Wnt inhibitor), leading to the formation of a catenin destruction complex.

Epithelial-mesenchymal metamorphosis and cancerous stem cells in pancreatic cancer are caused by *FOXM1* overexpression [111]. The pulmonary vasculature’s embryonic development depends on *FOXM1*, which is also involved in colon CSC proliferation and self-renewal. By controlling the primary stem cell regulator *SOX2*, *FOXM1* encourages the stemness of glioblastoma [112].

### 2.9. Members of the FOX Family as Drug Targets for Cancer Therapies

FOX proteins are climacteric controllers of various cellular and biological procedures. They have been proven to be crucial members of various pathways contributing to unhindered proliferation and tumorigenesis. Therefore, it is only just that drug therapies to block and/or alter such pathways be developed to treat different cancers. 

In many cancers, the *FOXM1* gene is exacerbated. Reactive oxygen species and oncogenic signaling pathways encourage its expression. It enhances cell migration, causes the epithelial-mesenchymal transition (EMT) composition in tumors, and creates a premetastatic niche in the distant metastasized organs. In addition, *FOXM1* promptly initializes genes that are implicated in numerous stages of metastasis. *FOXM1* is a vital player in the modulation of tumor magnification and metastasis. Thiostrepton slows the progression of laryngeal epidermoid carcinoma and breast cancer while inducing apoptosis [113]. 

Siomycin A was discovered to be a *FOXM1* inhibitor that suppressed *FOXM1’s* downriver gene targets, including Cdc25B, survivin, and CENPB. In pliable agar, siomycin A had the capacity to slow down the development of cells independent of anchoring. Furthermore, siomycin A selectively induced apoptosis in metamorphosized cells but not in unmetamorphosed cells of similar origin [114]. Treatment with a cell-permeable ARF26-44 peptide (FOXM1 inhibitor) decreased angiogenesis and tumor cell development and caused a consequential escalation in cell death within the HCC region but not in the surrounding healthy liver tissue. Several different human hepatoma cell lines were treated with the ARF peptide, and this caused the cells to die [115].

The *FOXO* family portrays a function in the progression of diabetes, cancer, and other human disorders. Several *FOXO1* direct-targeting therapeutic candidates are known to be under development, and some of them have already received patent protection. The intracellular location and functionality of *FOXO* proteins have been demonstrated to be affected by the small-molecule drugs D4476 and ETP-45658 [116]. Numerous malignancies have a negative correlation with *FOXO3A*. Adenovirus infection can effectively suppress the development of melanoma cells and cause a rapid loss of cell viability by overexpressing *FOXO3A*/triple mutant (TM). Adenovirus infection caused the cell cycle to be arrested and led to the overproduction of *FOXO3A*/TM in the carcinoma cell lines, according to cell cycle assessments [57].

When compared to healthy mammary tissues, the production of the *FOXQ1* protein was considerably increased in cases of basal-like breast cancer. It was also shown that the expression of the genes *FOXQ1* and Dachshund homolog 1 (DACH1) was inversely correlated in metastatic and non-metastatic breast cancers and other tumors. Additionally, *FOXQ1* helps cancer stem cells regenerate themselves. Diallyl Trisulfide’s targeting of *FOXQ1* prevents breast cancer stem cells from proliferating [117]. 

## 3. Conclusions

One of the fundamental reasons for targeting FOX proteins is that many FOX proteins act as pioneer factors. The fork-head gene family is critical in maintaining the proper functioning of cellular homeostasis through its involvement in various cellular pathways and processes. However, its importance in the growth and progression of breast cancer has piqued the interest of researchers. The alteration of certain subfamilies of the forkhead gene family leads to uncontrolled cell division, making it a target for drug therapies. Researchers are focusing on blocking the associated cellular pathways to prevent any cellular abnormalities. Additionally, the interaction between forkhead genes and microRNAs has been linked to cell cycle disruptions, and forkhead genes are also linked to the maintenance of the “stemness” of stem cells, which bodes well for future cancer research.

## Figures and Tables

**Figure 1 biomedicines-11-02159-f001:**
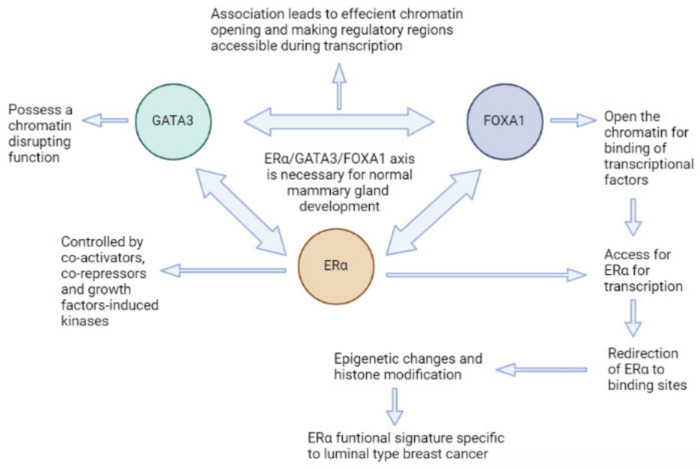
Erα-FOXA1-GATA3 Axis and its role in luminal type breast cancer development.

**Figure 2 biomedicines-11-02159-f002:**
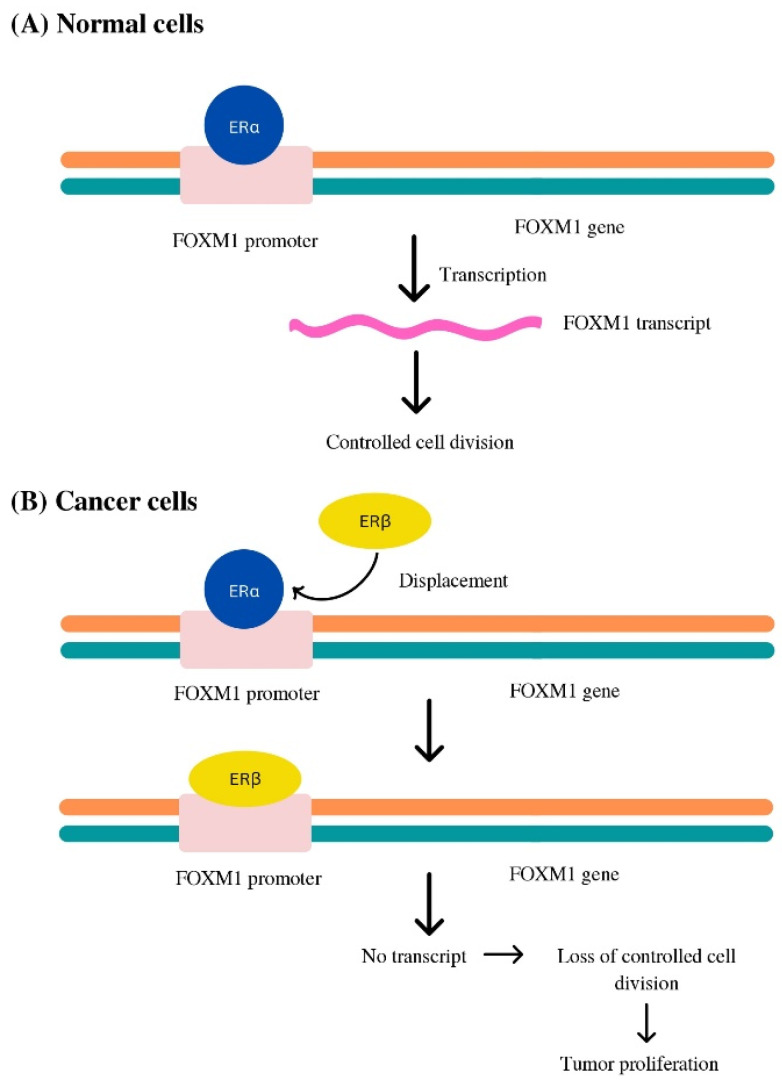
*FOXM1* transcriptome activity in (**A**) Normal cell (**B**) Luminal subtype of breast cancer cell.

**Figure 3 biomedicines-11-02159-f003:**
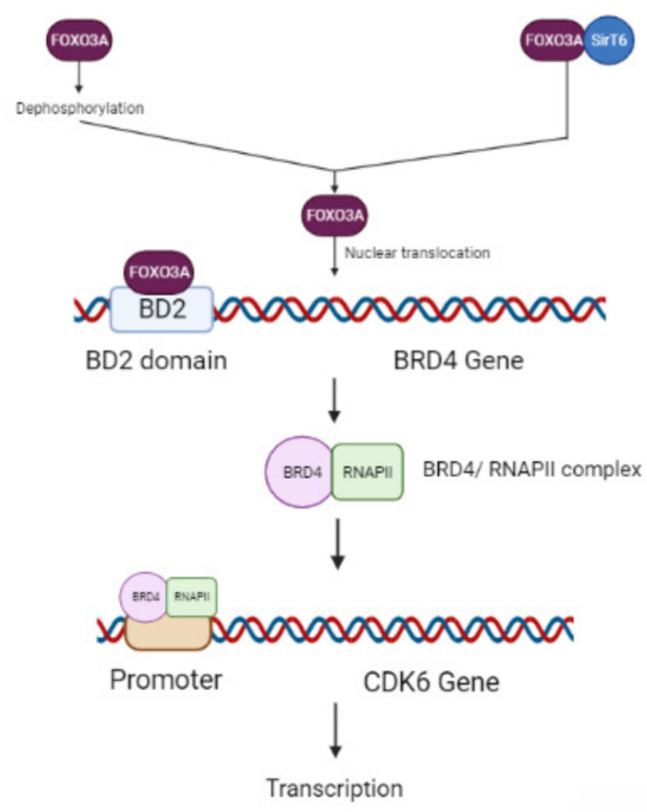
Akt inhibition therapy is used to treat luminal breast cancer. Resistance to therapy leads to the suppression of *BRD4/FOXO3A* or *CDK6* genes.

**Figure 4 biomedicines-11-02159-f004:**
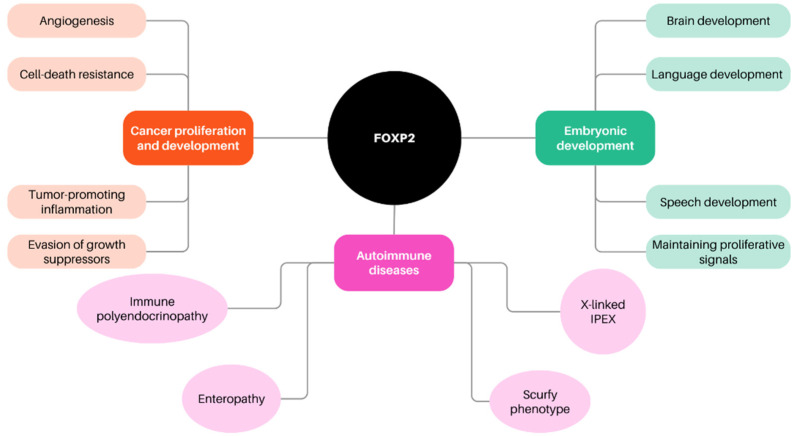
FOXP and its role in the various fundamental processes in the human body.

**Figure 5 biomedicines-11-02159-f005:**
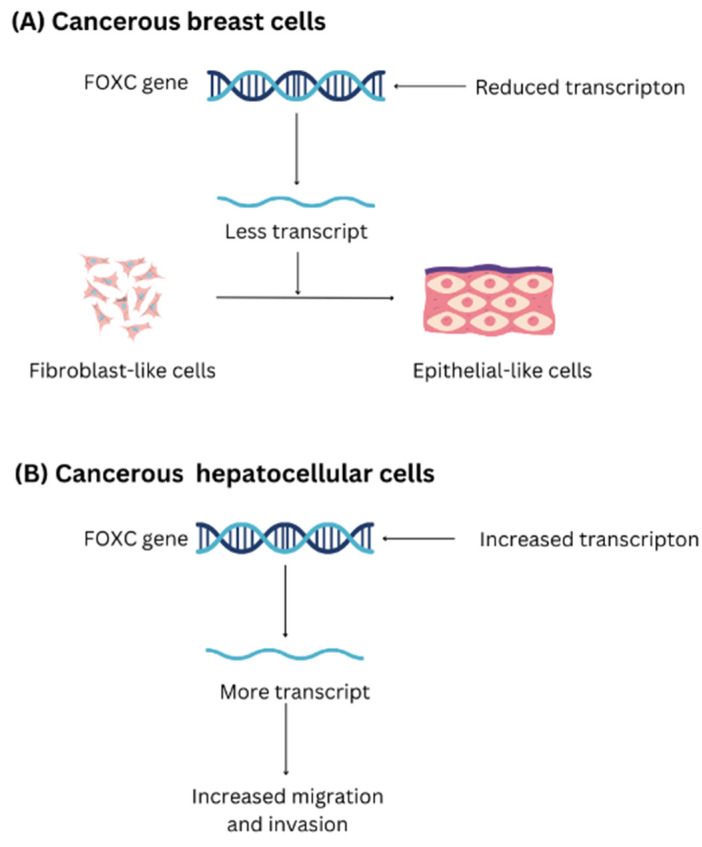
The anomalous behavior of the *FOXC* gene in the development and proliferation of (**A**) Breast cancer cell (**B**) hepatocellular cancer.

**Figure 6 biomedicines-11-02159-f006:**
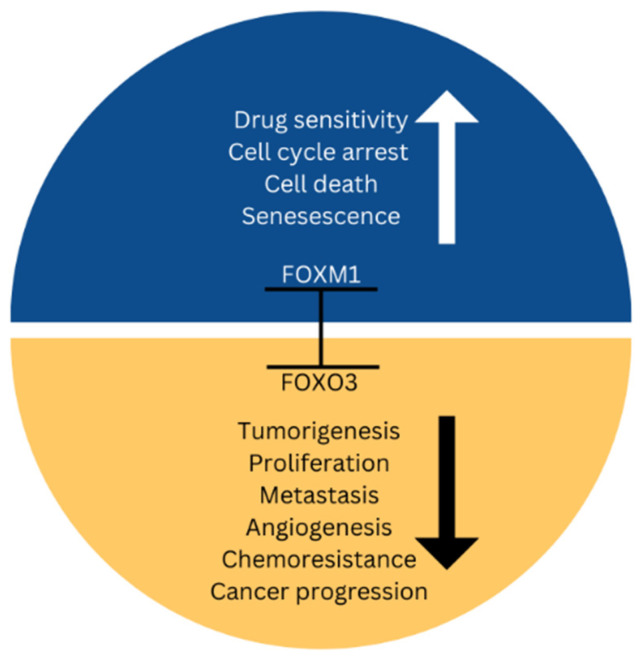
*FOXM1* and *FOXO* axis; the tumor suppressor *FOXO3* and the powerful oncogene *FOXM1* compete for activity and expression [79].

**Figure 7 biomedicines-11-02159-f007:**
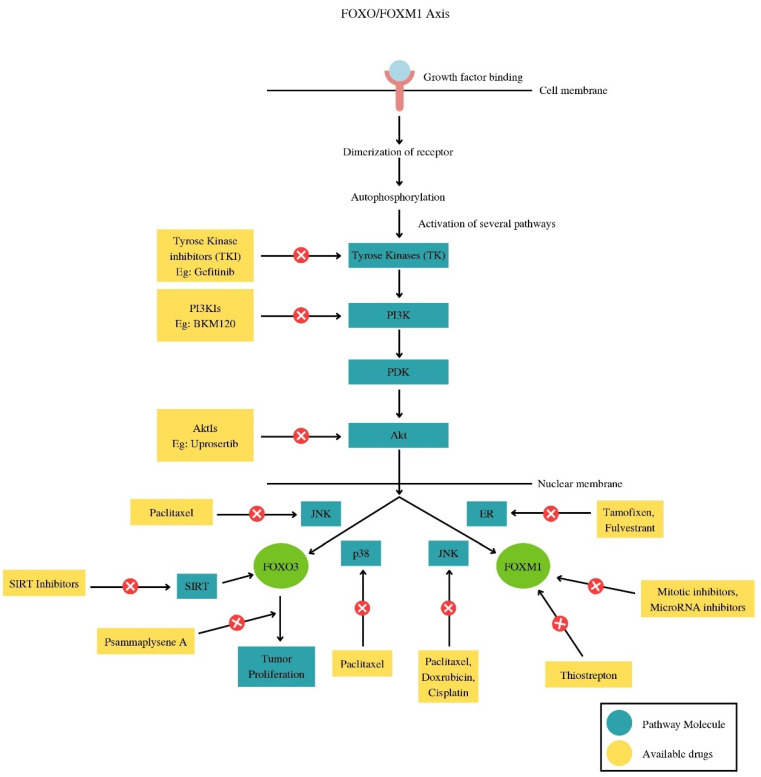
The drugs involved at every step in the different pathways related to the *FOXO/FOXM1* pathway.

**Table 1 biomedicines-11-02159-t001:** Direct and indirect associations of different *FOX* genes with cancer.

Cancer Hallmarks	*FOXA*	*FOXC*	*FOXM*	*FOXO*	*FOXP*
Invasion and metastasis	*FOXA1*, *FOXA2*, *FOXA3*	*FOXC1*, *FOXC2*	*FOXM1*	*FOXO1*, *FOXO3A*, *FOXO4*	*FOXP1*, *FOXP3*
Immune destruction	X	X	*X*	*FOXO1*	*FOXP1*, *FOXP3*, *FOXP4*
Cellular energetics	*FOXA2*	*FOXC2*	*FOXM1*	*FOXO1*, *FOXO6*, *FOXO3A*	X
Replicative immortality	*FOXA1*, *FOXA2*	*FOXC2*	*FOXM1*	*FOXO1*, *FOXO3A*, *FOXO4*	X
Evading growth suppressors	*FOXA1*, *FOXA2*	*X*	*FOXM1*	*FOXO1*, *FOXO3A*, *FOXO4*	*FOXP1*, *FOXP3*
Genome stability and mutation	*FOXA1*, *FOXA2*, *FOXA3*	*FOXC1*, *FOXC2*	*FOXM1*	*FOXO1*, *FOXO6*, *FOXO3A*, *FOXO4*	*FOXP1*, *FOXP2*, *FOXP3*, *FOXP4*
Inducing angiogenesis	*FOXA1*, *FOXA2*	*FOXC1*, *FOXC2*	*FOXM1*	*FOXO1*, *FOXO3A*	*FOXP1*, *FOXP3*
Resisting cell death	*FOXA1*, *FOXA2*, *FOXA3*	*X*	*FOXM1*	*FOXO1*, *FOXO6*, *FOXO3A*, *FOXO4*	*FOXP1*, *FOXP3*
Sustaining proliferative signaling	*FOXA1*, *FOXA2*	*FOXC1*, *FOXC2*	*FOXM1*	*FOXO1*, *FOXO6*, *FOXO3A*, *FOXO4*	*FOXP1*, *FOXP3*
Tumor proliferative inflammation	*FOXA1*, *FOXA3*	*FOXC1*	*FOXM1*	*FOXO1*, *FOXO6*, *FOXO3A*, *FOXO4*	*FOXP3*

**Table 2 biomedicines-11-02159-t002:** Interaction of various miRNAs with various forkhead-box genes and their association with different cancers.

miRNA Class	Interaction with Forkhead Box	Associated Cancer	References
miR-342	*FOXM1*, *FOXQ1*	Colorectal cancer	[87]
miR-204	*FOXM1*	Esophageal cancer	[88]
miR-422a	*FOXQ1*, *FOXG1*, *FOXE1*	Liver cancer	[91]
miRNAs	*FOXO3A*	HER2 Positive breast cancer	[92,93]
miR200b-5p	*FOXM1*	Breast cancer	[97]
miR-186-5p	*FOXM1*	Breast cancer	[97]
miRNA-937	*FOXQ1*	Stomach, lung, breast cancer	[98,99,100]

## Data Availability

Not applicable.

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
