# Peer review of "Role of Fork-Head Box Genes in Breast Cancer: From Drug Resistance to Therapeutic Targets"

_biomedicines, 2023, doi:10.3390/biomedicines11082159_

Round 1

Reviewer 1 Report

This review is covering interesting subjects and I think it will attract some interest from readers. However, some improvements are needed.

1.                   For fork-head box genes, the degree of sequence similarity of the individual genes in the constituent genes should be shown graphically.

2.                   It should be analyzed whether there is a correlation between the similarity of the sequence and the function.

3.                   There is little description of the nucleotide sequences that each member of the fork-head box genes recognizes and binds. The reader will want to know how much diversity there is in the recognition sequences.

Author Response

Response to Reviewer 1

  1. The degree of the sequence similarities of FORK-head family genes has been already published graphically by Lam et al., 2013 in Nature Reviews Cancer. We have incorporated the information and cited the reference in the MS. Highlighted from Line number 45 to 64.

Reference number [2]: Lam E.W.F., Brosens J.J., Gomes A.R., Koo C.-Y. Forkhead box proteins: Tuning forks for transcriptional harmony. Nat. Rev. Cancer. 2013;13:482. doi: 10.1038/nrc3539.

  1. The MS is updated and highlighted accordingly. Highlighted from Line number 45 to 64.
  2. The MS is updated. Highlighted from Line number 45 to 64.

Reviewer 2 Report

The authors concluded various fork head genes and how they play a role in different types of cancer. The paper highlights the complex interactions between  fork head genes and their impact on tissue homeostasis and cancer. The following are my comments and critique:

1. The authors need to make a schematic diagram to clearly show the roles of Forkhead Box P and Forkhead Box C in breast cancer.

2. The authors are suggested to list the effect or function of FOX proteins in breast cancer drug therapy.

3. The authors are suggested to list the miRNAs which regulate FOX proteins and introduce the concrete correlations between them in the table.

4. The mechanisms about how FOX family function in stem cells need to be introduced more.

Grammatical errors and undefined abbreviations are present in the manuscript that needs correction. Minor typing mistakes should be corrected. The authors get editing help from someone with full professional proficiency in English.

Author Response

Response to reviewer#2

  1. The schematic diagrams are included in MS (Figure4&5)
  2. The drug therapy in breast cancer related to FOX proteins in illustrated in figure7 and in para2.8
  3. The list of miRNAs is included in Table 2 in MS.
  4. The mechanism via various pathway is highlighted in MS. Line no 428-33 and 450-53

The grammatical mistakes have been updated and MS is being proofread by Grammarly premium.

Reviewer 3 Report

Reviewer’s comments to Author:

1.     The focus of this article is on the discussion: Role of Fork-head box genes in breast Cancer: From Drug Resistance to Therapeutic Targets.

2. In “Role of various FOX-head boxes in tumor proliferation and progression” sections, there are detailed references and discussions.

a.     Is it clear in the conclusion that some factors in FOX-head genes are involved in hormone regulation, immune system modulation, and disease progression through their regulation of the epithelial–mesenchymal transition? Those FOX-head factors can influence cancer development, progression, plays an absolutely important role in the process of metastasis, and drug resistance?

b.  Previous researchers have proposed that the drug resistance mechanisms include the development of resistance to apoptosis, increased DNA damage repair efficiency, elimination of the chemotherapeutic drugs by increased efflux, and drug inactivation, et. Factors also have the same effect in breast cancer?.

c.      On Functions of FOX-head factors in cancer: discuss the roles of FOX factors in the epithelial–mesenchymal transition (EMT), hormone signaling, drug resistance, metabolism, and immune system regulation and their functions as pioneering factors, whether you can make suggestions What are the absolute key factors in the targeted therapy of cancer in the future? Is there any advantage compared with the current point of view of targeted therapy for breast cancer?

d. It is suggested to cite more evidence-based medicine discussing Cancer Metabolism, Evasion and Regulation of the Immune System.

e. Are there any suggestions for targeted therapy for triple-negative breast cancer?

3.  A brief statement "One of the fundamental reasons for targeting FOX proteins is that many FOX proteins act as pioneer factors" can be added to the conclusion. What are some of the more important proposals for future directions and pioneer factors?

4.  Please briefly describe the important learning objectives of this Review article?

Author Response

Reviewer 3

1 a) Certainly, the FOX head proteins are involved in hormone regulation, immune system modulation and disease progression leading to cancer development. The factors and specific genes involved are being described in MS. Also, the individual gene involved in all these events have been explained in table 1.

  1. b) yes, in resistant breast cancer cell line the same observations were noticed.
  2. c) In the MS it is clearly indicated that FOXM1 controls cancer mitosis and EMT. And also, the role of each factor is explored in elaboration in the MS. The targeted therapies in breast cancer are huge area to explore although we have tried to come up with recent research exploring FOX head genes in Breast cancer therapies.
  3. d) we have tried to mention drug resistance related to FOX-head genes in breast cancer. The drugs involved are being covered in the MS. I appreciate your suggestion but this would include entirely new and huge topic for reviewing which we will consider for further publications.
  4. e) FOXA2 is known to be targeted for TNBC patients. The details are highlighted in MS. Line number 127-133.

3) Changes are made in MS and highlighted. Line no. 497-98

4) the learning objectives of review article is already mentioned in abstract section of MS.

Round 2

Reviewer 1 Report

Thank you for revising your manuscript.

I am happy to tell you that your manuscript is improved and should be  accepted in the Journal.

Reviewer 2 Report

In the revised article, the authors modified the manuscript and figures referred to the comments, and answered the questions comprehensively.